# Can digital economy improve employment structure?—Mediating effect based on a spatial Durbin model

Yang Lu[☯], Lu Lu Zhou[iD]*[☯]

School of Economics, Jiangxi University of Finance and Economics, Nanchang, Jiangxi, China

☯ These authors contributed equally to this work.
* zll13972023@163.com

**Data Availability Statement:** All relevant data are within the manuscript and its Supporting information files.

## Abstract

Based on the panel data of 283 prefecture-level cities in China from 2011 to 2019, this study constructed an index measurement system of digital economy, economic agglomeration, innovation and entrepreneurship, and employment structure. The index of digital economy was developed by entropy weight method, and the double-fixed spatial Durbin model was constructed based on the intermediary effect from the spatial perspective to determine the direct effect, indirect effect, and total effect of the digital economy, economic agglomeration degree, and innovation and entrepreneurship on employment structure. The results indicated a significant spatial correlation between the three aspects, i.e., digital economy can significantly optimize the employment structure, with an evident spillover effect. The mechanism analysis revealed that the level of innovation and entrepreneurship poses a stronger intermediary effect than the degree of economic agglomeration, and the digital economy in the eastern region directly impacts the urban employment structure; however, the influence of digital economy on the employment structure is significantly higher in small- and medium-sized cities than in large-sized cities.

## Introduction

Employment is the foundation of livelihood as it provides the most significant means of livelihood, support, and development, thereby ensuring social stability. According to the 14th Five-Year Plan (2021–2025), China's urban economic development is encountering complex structural contradictions, which is propelling an economic downturn, with several enterprises announcing layoffs and further aggravating the employment scenario. To address this arduous task, employment opportunities need to be extended while promoting reemployment. With the continuous development of digital and network technology, digital economy has profoundly impacted the production and employment structure in China. In principle, digital economy is characterized by high efficiency, intelligence, and flexibility, which promotes industrial transformation and upgradation for accelerating economic development. However, digital economy has disrupted conventional employment models, especially in traditional industries, where workflows are undergoing continuous automation that require minimal

**Funding:** 1. Fundamental Project: Jiangxi Provincial Social Science Foundation Project "Research on the Impact of Digital Economy Development on Employment Structure and Quality in Jiangxi Province and Countermeasures" (Grant No. 23YJ55D). 2. Jiangxi Province University Humanities and Social Sciences Research Project "Research on the Dynamic Mechanism and Countermeasures of Industrial Digitalization to Promote the High-quality Development of Jiangxi's Manufacturing Industry" (Grant No.JJ22218).

**Competing interests:** The authors have declared that no competing interests exist.

human intervention. Although digital transformation has affected the employment structure of conventional industries, it has ushered new employment opportunities and employment methods. Thus, this research explores the impact of digital economy on the employment structure and analyzes the factors governing the associated phenomena, which is expected to promote China's economic development and upgrade its employment level.

## Literature review

The existing literature is focused on determining the effects of digital economy on the employment structure of various industries and the realization of this impact through transformations in the overall employment scale of the industry. In terms of the substitution effect of the digital economy, Anke et al. (2019) reported that in the short term, digital technology will replace a greater proportion of manufacturing employment than creating employment in the service sector, and this trend will magnify and exacerbate the polarization of income [1]. Acemoglu and Restrepo (2020) studied the U.S. labor market and unveiled that adopting a single robot per 1,000 workers can reduce the employment-to-population ratio by 0.2% [2] Ziru et al. (2021) suggested that both structural and frictional unemployment are inevitable in the short term [3]. Zhang et al. (2022) collected data from 30 provinces in China over a six-year period and concluded that technological innovation has disrupted the structural imbalance in digital economy's labor market, resulting in employment polarization [4]. Eder et al. (2022) explored the impact of digital technology on the size of employment in Austria and demonstrated that new technological elements will increase the employment of high-skilled and low-skilled labor and reduce the employment of medium-skilled labor, thereby polarizing the employment structure [5]. Lu (2023) finds that the overall impact of the digital economy on local technological innovation is not significant based on the panel data of 30 provinces in China from 2006 to 2018 [6]. In terms of the creative effects of the digital economy, Graetz and Michaels (2018) revealed that the digitization of enterprises in the industrial sector reduces the share of employment in the sector and promotes employment in high-end services [7]. Amuso and Poletti et al. (2019) reported that the rapid development of advanced-generation information technology has ushered new forms of economic development and employment methods, which have created additional jobs and employment opportunities [8]. Ballestar (2020) demonstrated the complementary relationship between employment and the development of digital economy, even highlighting recent improvements in the employment structure [9]. Dennis (2021) noted that digital economy has prompted firms to reduce vacancies, which has forced job seekers to attenuate their search for jobs in the automated sector [10]. However, the decline in manufacturing employment can be compensated by an increase in service employment. Wang et al. (2022) used a dynamic spatial Durbin model to study the digital economy index of 30 provinces in China and found that the accelerated development of digital technology will optimize the employment structure and significantly promote the development of green economy [11]. Du et al. (2022) believe that the accelerated development of digital technology has extended employee mobility and created new segments of high-quality job [12]. Based on China's provincial panel data for 2011–2019, Zhao et al. (2023) posited that the development of the digital economy can effectively curb employment polarization and achieve significant gains [13]. Bu (2023) studied the use of spatial Durbin models to measure the industrial structural transformation index of 30 provinces in China and indicated that the development of digital finance can significantly promote the upgrading of local industrial structure [14].

Based on the collation and analysis of the existing literature, the research on the impact assessment of digital economy towards altering the employment structure poses the following shortcomings: first, the majority of the existing literature applied common benchmark

regression models to assess the impact of digital economy on the variations in employment structure, and did not construct the models from the geospatial perspective to explore its mechanism; second, most prior research analyzed the impact of digital economy on the employment structure based on provincial-level data, which is not extensive and lacks comprehensiveness. To this end, the present study extends the following contributions. First, the intermediary effect is explored based on the spatial perspective to examine the spillover effect of the spatial Durbin model, and the impact of digital economy development on China's employment structure is discussed along with its influencing mechanism, which provides new insights for foundational research. Second, this research analyzed data acquired from 283 cities in China between 2011 and 2019 to characterize the relationship between digital economy development and employment structure transformation in China.

## Theoretical analysis and research hypothesis

### Mechanism of digital economy altering the employment structure

Digitalization, also known as the fourth industrial revolution, has gradually become integrated into all aspects of our lives, resulting in qualitative and quantitative changes in the employment structure of the labor force [15]. Although it has considerably transformed the carrier, form, and skill demand of employment, its impact on employment is two-sided—employment creation as well as substitutio. The employment substitution effect of digital economy has resulted in considerable employment loss among the labor force engaged in manufacturing industries. Similarly, low-skilled jobs have been replaced by complex artificial intelligence and big data, which has improved the productivity of manufacturing enterprises. However, this replacement of labor by digitalization is endangering the livelihood of numerous working professionals. In contrast, digital development has spawned new business models, reorganized production processes, and created new benefits and value addition processes [16]. Therefore, in the long term, the employment creation effect of the digital economy is expected to outweigh its substitution effect. With further development of digital technology, personal productivity will improve remarkably in future and the job market is expected to become more flexible. Consequently, the employment of the labor force will gradually shift from the primary and secondary industries to the tertiary industry, thereby increasing the proportion of employment in the service industry. Therefore, this research considers the following hypothesis:

Hypothesis H1: The development of the digital economy exerts a positive effect on optimizing the employment structure.

### Intermediary effect mechanism of digital economy affecting the employment structure

The digital economy can indirectly affect the employment structure by promoting economic agglomeration, driving industrial innovation, and stimulating employment.

First, the level of digitalization among enterprises is relatively high in large cities with high-level economic development and mature infrastructure. Moreover, the development of the digital economy has diversified business operations and created more flexible forms of employment, which can attract high-skilled labor and high-quality innovation. In principle, capital promotes the accumulation of urban talents and industries, significantly affects scale and economic agglomeration, and promotes alterations in the urban employment structure [17]. Second, as a primary factor of production, the continuous development and management of data will promote the structural transformation of urban industries and services. This can

consequently increase jobs as well as promote labor flow between various industries, absorb the surplus labor force replaced by digitalization, ease the employment pressure, and optimize the regional employment structure. Finally, the innovations resulting from the continuous development of the digital economy can aid enterprises to weaken their existing information asymmetry, reduce their operational costs, increase their market value/share by providing competitive advantage, and promote their digital transformation. The renewal and upgradation of enterprises through scientific and technological achievements will require corresponding data, human resources, and capital to the region, thereby inducing the regional innovation effect. Based on the aforementioned analysis, the second hypothesis of this research is postulated as follows:

Hypothesis H2: The development of digital economy can impact China's employment structure *via* two mechanisms, namely, the effect of economic agglomeration and the effect of innovation and entrepreneurship.

## Regional heterogeneity of the impact of digital economy on employment structure

As the endowment of resources varies considerably across the geographical environment in China, several scholars have accounted for these geospatial variations while assessing the impact of digital economy on the employment structure. Hat et al. (2020) adopted the Austrian case study to report that only a few urban areas and small towns encounter the issue of disappearing jobs, and labor substitutability is relatively higher in rural areas than in urban cities [18]. Zhang et al. (2021) reported that digital economy has promoted the development of China's industrial structure, and this growth is characterized by regional heterogeneity [19]. Upon comparing the provincial and nonprovincial cities, the demographic heterogeneity is influenced by the regional division as well as the total population of the city. Mao et al. (2023) pointed out that there are significant regional differences in the development of China's digital economy [20]. Thus, owing to varying degrees of development of the digital economy across various regions, the degree of its impact on the employment structure varies as well. Accordingly, the third hypothesis of this research is stated as follows:

Hypothesis H3: The impact of digital economic development on China's employment structure is characterized by regional heterogeneity.

## Study design

### Empirical model setting

**Spatial correlation test.** Based on prior research, this study employed Moran's I (Index) method to evaluate the spatial correlation between the digital economy and employment structure. The Moran's I statistic for the spatial correlation test can be expressed as follows:

$$\text{Moran's } I = \frac{\sum_{i=1}^{n} \sum_{j=1}^{n} w_{ij}(Y_i - \bar{Y})\left(Y_j - \bar{Y}\right)}{S^2 \sum_{i=1}^{n} \sum_{j=1}^{n} w_{ij}}, \tag{1}$$

where $S^2$ represents the sample variance, $\bar{Y}$ represents the sample mean, $Y_i$ and $Y_j$ represent the observations for the $i$-th and $j$-th regions, respectively, and $n$ represents the number of cities, in this case 283, $w_{ij}$ denotes a spatial weight matrix. Based on the significance of Moran's I, the positive and negative values, size, and spatial correlation can be assessed comprehensively.

**Spatial measurement model setting.** The spatial measurement model conducts various tests to determine the specific model suitable for the present data. Accordingly, this study applied the LM test(Lagrange Multiplier Test) and Hausman test [21], combined with the significance test, which demonstrated the stronger effect of the double-fixed effect model than the spatial- and time-fixed effects. Finally, based on the obtained experimental results, a spatial Durbin model was constructed considering the double-fixed effects, expressed as follows:

$$LS_{it} = \beta_0 + \rho \sum_j w_{ij} LS_{jt} + \beta_1 DE_{it} + \delta_1 \sum_j w_{ij} DE_{jt} + \beta_2 Z_{it} + \delta_2 \sum_j w_{ij} Z_{jt} + \mu_i + \gamma_t + \varepsilon_{it}, \quad (2)$$

where $i$ and $j$ denotes the regions, $t$ represents the years, and labor structure (LS) represents the employment structure of the labor force at the industry level. DE represents the level of the digital economy, $w_{ij}$ indicates the spatial weight matrix, $Z_{it}$ denotes the control variable of the model, $\mu_i$ and $\gamma_t$ represent the spatial fixed effect, respectively, $\gamma_t$ and $\varepsilon_{it}$ denotes the random disturbance term.

**Mediation test setting.** Whether economic agglomeration and innovation and entrepreneurship can be used as intermediary variables, the size of the mechanism between the two intermediary variables, the role of the path and the intermediary effect exist? Referring to the conclusions of Wen et al. (2004) [22], the intermediary effect mechanism was tested and analyzed from the spatial perspective based on the regression of the spatial Durbin model. The specific model settings are stated as follows:

$$M_{it} = \theta_0 + \rho_M \sum_j w_{ij} M_{jt} + \theta_1 de_{it} + \phi_1 \sum_j w_{ij} de_{jt} + \theta_2 Z_{it} + \phi_2 \sum_j w_{ij} Z_{jt} + \mu_{it} + \gamma_t + \varepsilon_{it}, \quad (3)$$

$$LE_{it} = \beta_0' + \rho' \sum_j w_{ij} LE_{jt} + \beta_1' DE_{it} + \delta_1' \sum_j w_{ij} DE_{jt} + \beta_2' M_{it} + \delta_2' \sum_j w_{ij} M_{jt} + \beta_3' Z_{it}$$
$$+ \delta_3' \sum_j w_{ij} Z_{jt} + \mu_i + \gamma_t + \varepsilon_{it}, \quad (4)$$

where $M_{it}$ denotes the intermediary variable, indicating the degree of economic concentration and the level of innovation and entrepreneurship; the remaining variables and symbols express the same meaning as that in Eq (2).

As depicted in Fig 1, the intermediary effect test refer to the specific test steps reported in the existing literature [23], because this research presumes two intermediary variables and comparatively analyzes the strength of the two intermediary mechanisms.

## Variable selection

**Explained variables.** According to a previous study [24], the employment structure (LS) was considered the explanatory variable, which was primarily determined according to the proportion of the population employed in each industry or the proportion of GDP contributed by three industries, disregarding the movement of agricultural and non-agricultural employment. However, the employment structure thus described contains certain errors. Yang (2022) measured the employment structure at two levels [25]—(1) industry level: the manufacturing industry is significantly affected by the digital economy owing to the characteristics of the labor force, and studying the fluctuations of the employment population in the manufacturing industry deepens the understanding of the employment structure. Therefore, the present research measured the employment structure [26] by evaluating the proportion of the total population employed in manufacturing industries in each city. (2) Industry Excess Level: during the development of DE, the labor force employed in the primary and secondary industries

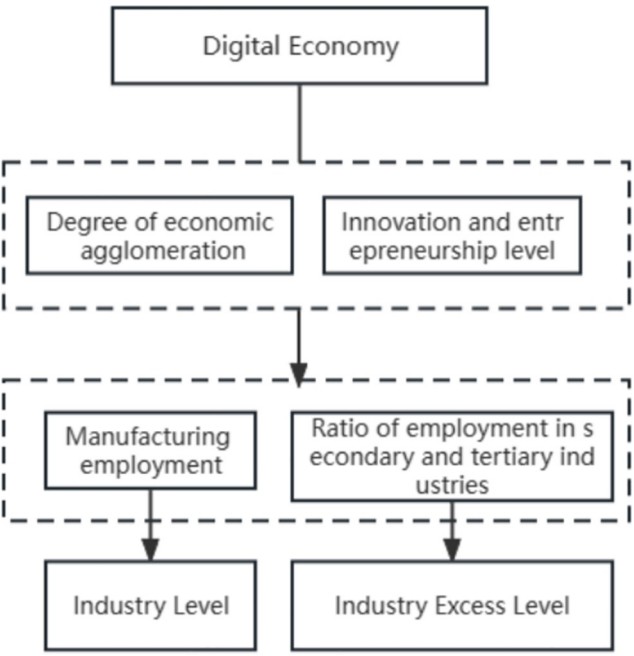

**Fig 1. Schematic of mediation mechanism.**

progressively transitions to tertiary industries. Thus, this study evaluated the ratio of the number of employees in the tertiary industry to that in the secondary industry.

**Core explanatory variables.** Herein, the development of DE is considered the core explanatory variable. Combined with the existing literature research, the entropy right-Topsis method was employed to construct a comprehensive evaluation index system of nine positive indicators in three dimensions of digital foundation and digital application. As shown in Table 1. The factors affecting the DE development in China were analyzed based on these evaluation indices (owing to space limitations, the specific calculation process is not presented here).

**Control variables.** To a certain extent, China's employment structure is affected by additional factors. According to prior research, the following control variables were selected. First,

**Table 1. Indicator system for explanatory variables.**

| Main Indicators | Level 1 indicators | Secondary indicators | Indicator Properties |
|---|---|---|---|
| Development of Digital Economy | Digital Foundation | Science and technology expenditure in government Fiscal expenditure (10,000 Yuan) | + |
| | | Information transmission, software, and information technology services for urban employment (in 10,000) | + |
| | Digital Applications | Mobile phone penetration rate (units/per 100 people) | + |
| | | Mobile users (million households) | + |
| | | Number of users with access to Internet broadband (in 1000) | + |
| | Scale of Digital Economy | Peking University Inclusive Financial Index | + |
| | | Total telecommunications services ($million) | + |
| | | Postal business revenue ($million) | + |
| | | Total retail sales of consumer goods ($million) | + |

the industrial structure (STR). Zhao (2022) concluded that the improvement of the industrial structure will propel the transfer of labor force to realize the optimization and adjustment of the employment structure. Therefore, the ratio of regional industrial value added to local GDP was regarded as an index. This ratio reflects the proportion of industry in the regional economy. When the ratio of industrial added value to GDP is high, it indicates that the industrial sector in the region is relatively developed, and thus the economic development tends to be good [27]. Second, the level of regional economic development (RED) was referred from the existing literature [28] to analyze the per capita GDP of each province. Generally, per capita GDP can comprehensively reflect the degree of economic development, and the increase in the degree of RED increases the variations in the employment structure. Third, the capacity of labor supply in technology (CST) is predominantly composed by students, and their preference will eventually alter the employment structure. Therefore, Yao et al. (2020) [29] considered the number of students graduating from colleges and universities as an index. Fourth, the government financial support (GOV) received through the government's fiscal expenditure can affect industrial development, which correspondingly alters the employment structure. Thus, this research evaluated the ratio of the local government's general fiscal expenditure to local GDP [11].

**Intermediary variable.** Degree of economic agglomeration (Density): Generally, areas with a higher level economic development and mature infrastructure attract a larger volume of economic inflows. Therefore, according to Henderson(2021) [30], this study considered the economic density of urban units as the index and presumed the ratio of GDP of each city to the urban land area for calculation.

Innovation and entrepreneurship level (Inn): Based on Zeng (2022) [31], this study selected the innovation and entrepreneurship index of a city, i.e., the degree of innovation in an enterprise, as an intermediary variable with the positive index.

## Research samples and data sources

Based on the availability and completeness of the data, this research selected 283 cities across all prefectures in China from 2011 to 2019 as the research sample, comprising a total of 2547 observations. Excluding the Digital Inclusive Finance Index and the Innovation and Entrepreneurship Index from the "Digital Inclusive Finance" of Peking University and the Enterprise Big Data Research Center, the original data of the remaining variables were derived from the 2012–2020 China Urban Statistical Yearbook and the statistical yearbooks of each city. Certain indicators were further evaluated based on the original data. The individual missing values were imputed [32] by manually consulting the statistical bulletin of the national economic and social development of each prefecture-level city or through linear interpolation. In addition, to avoid the influence of outliers, the logarithm of each variable was considered. The descriptive statistics of the variables are listed in Table 2.

## Empirical analysis

### Spatial correlation analysis

As discussed earlier, the Moran's I (Index) method [33] was employed to test the correlation between the employment structure, level of DE development, degree of economic agglomeration, and level of innovation and entrepreneurship. The results are listed in Table 3.

As listed in Table 3, during the nine years (2011 to 2019) covered by the sample data, the global Moran's I values of the employment structure, level of digital economic development, degree of economic agglomeration, and the level of innovation and entrepreneurship in each region were significantly positive, indicating the spatial correlation between these variables,

**Table 2. Descriptive Statistics of the present sample data.**

| Variable name | Variable symbol | Number of samples | Mean | Standard deviation | Minimum | Maximum |
|---|---|---|---|---|---|---|
| industry level | is | 2547 | 0.236 | 0.136 | 0.006 | 0.813 |
| Industry Excess Level | ies | 2547 | 0.987 | 0.660 | 0.047 | 5.485 |
| Development level of digital economy | de | 2547 | 0.057 | 0.083 | 0.002 | 0.921 |
| Industrial structure | str | 2547 | 0.391 | 0.124 | 0.032 | 0.871 |
| Government financial support | gov | 2547 | 0.202 | 0.103 | 0.009 | 0.916 |
| Level of Regional Economic Development (10,000 Yuan) | red | 2547 | 56116.781 | 130893.08 | 6457 | 6421762 |
| Labor technology supply capacity (person) | cst | 2547 | 94573.621 | 166915.71 | 225 | 1152994 |
| Degree of economic agglomeration | den | 2547 | 1.635 | 2.261 | 0.006 | 22.383 |
| Innovation and entrepreneurship level | inn | 2547 | 4.352 | 0.201 | 3.177 | 4.605 |

i.e., these variables are closely related to the level of development in the neighboring regions. The analysis results revealed that Moran's I of the level of employment structure and the level of DE development fluctuated marginally within a certain range during this period, indicating that the spatial correlation between these variables varied slightly, but overall, remained relatively stable. In contrast, Moran's I of the degree of economic agglomeration and the level of innovation and entrepreneurship fluctuated considerably, with an overall upward trend attaining peaks in 2019. Thus, the degree of economic agglomeration and the spatial correlation of innovation and entrepreneurship between these regions increased gradually. These findings prominently imply the spatial correlation between the variables, and accordingly, a spatial econometric model with spatial effects should be selected.

## Spatial econometric regression analysis

**Regression analysis of space Durbin model.** A series of tests must be conducted to select the spatial econometric model suitable for the current analysis. First, the panel data comprising the samples from 283 cities were set for the fixed-effect and random-effect tests [34], and based on the Hausman test, the fixed effect model was selected. In the second step, an LR test (Likelihood Ratio Test) was performed with an LR statistics of 28.71 and a $p$-value of 0.000, which strongly rejected the original hypothesis of the single fixed effects. Therefore, the spatial Durbin model was selected under the double-fixed effect to test the spatial effect of the DE on the employment structure. The model estimation results are listed in Table 4.

**Table 3. Global Moran's I (Index) values.**

| Year | Industry Level | | Industry Excess Level | | Development Level of Digital Economy | | Degree of Economic Agglomeration | | Innovation and Entrepreneurship Level | |
|---|---|---|---|---|---|---|---|---|---|---|
| | Moran's I | $p$-value | Moran's I | $p$-value | Moran's I | $p$-value | Moran's I | $p$-value | Moran's I | $p$-value |
| 2011 | 0.070 | 0.000 | 0.025 | 0.000 | 0.050 | 0.000 | 0.130 | 0.000 | 0.109 | 0.000 |
| 2012 | 0.080 | 0.000 | 0.027 | 0.000 | 0.045 | 0.000 | 0.128 | 0.000 | 0.083 | 0.000 |
| 2013 | 0.106 | 0.000 | 0.034 | 0.000 | 0.047 | 0.000 | 0.096 | 0.000 | 0.096 | 0.000 |
| 2014 | 0.092 | 0.000 | 0.048 | 0.000 | 0.044 | 0.000 | 0.121 | 0.000 | 0.061 | 0.000 |
| 2015 | 0.091 | 0.000 | 0.051 | 0.000 | 0.039 | 0.000 | 0.119 | 0.000 | 0.095 | 0.000 |
| 2016 | 0.095 | 0.000 | 0.055 | 0.000 | 0.038 | 0.000 | 0.127 | 0.000 | 0.105 | 0.000 |
| 2017 | 0.098 | 0.000 | 0.063 | 0.000 | 0.029 | 0.000 | 0.130 | 0.000 | 0.126 | 0.000 |
| 2018 | 0.117 | 0.000 | 0.073 | 0.000 | 0.033 | 0.000 | 0.135 | 0.000 | 0.137 | 0.000 |
| 2019 | 0.119 | 0.000 | 0.070 | 0.000 | 0.031 | 0.000 | 0.144 | 0.000 | 0.139 | 0.000 |

**Table 4. Regression results of spatial Durbin model.**

| Variable name | Model 1 | Variable name | Model 2 |
|---|---|---|---|
| | lnis | | lnies |
| lnde | 0.048** (2.48) | lnde | 0.130 *** (6.57) |
| lnstr | 0.466 *** (15.37) | lnstr | 0.571 *** (20.82) |
| lngov | −0.480 *** (−13.9) | lngov | −0.557 *** (−18.11) |
| lnred | 0.042 * (1.68) | lnred | 0.074 *** (3.15) |
| lncst | 0.040 *** (3.50) | lncst | −0.084 *** (−8.02) |
| W.lnde | 2.422 *** (8.96) | W.lnde | 0.085 *** (3.87) |
| W.lnstr | 0.741** (2.54) | W.lnstr | 1.199 *** (4.02) |
| W.lngov | −0.909 *** (−2.73) | W.lngov | 1.068 *** (3.51) |
| Wlnred | −1.602 *** (−8.34) | Wlnred | 0.099 (0.68) |
| W.lncst | −0.884 *** (−6.89) | W.lncst | −0.218** (−1.98) |
| Province fixed effect | | Yes | |
| Year fixed effect | | Yes | |
| Region | | 283 | |
| Number of observations | | 2547 | |

Note: Significance level is

*$p < 0.05$,

**$p < 0.01$,

***$p < 0.001$.

Model 1 uses an adjacent weight matrix W01. In the regression model, the logarithmic values of the variables are used to attenuate the effect of heteroskedasticity.

As observed for models 1 and 2 in Table 4, the regression coefficients of the core explanatory variable "lnde" and the variable "W.lnde" were positive at the 1% significance level. Thus, the development of DE poses a significant optimization effect on the employment structure at both levels. The 1-fold growth of DE will promote the growth of employment in the secondary and manufacturing industries by 13% and 4.8% compared to the tertiary industry. Therefore, DE can result in a high-tech employment structure in manufacturing industries. However, the results indicated a deviation from the reality, i.e., the service industry is more likely to drive employment. This is because DE development promotes innovation in information technology, which consequently facilitates digital transformation, upgradation, and intelligence-oriented development of classical industries. Therefore, the demand for digital industry-related industries is increasing continually, which increases the labor requirement in these industries. For example, in the field of DE, the emerging e-commerce and logistics industries demand considerable human and material resources, and these industries are closely related to secondary industries such as traditional manufacturing and industry, which were one of the major sources of labor in these industries. In addition, the emergence of the DE has driven the development of several related industries such as finance, telecommunications, and information technology. These industries require an extensive number of technical, management, and marketing talents, which is expected to stimulate the labor demand of the secondary industries. Therefore, with the continuous development of the DE, the demand for personnel in the secondary industries will increase as well, which will motivate numerous professionals to enter the development track of digital transformation and intelligence.

The regression coefficient of the control variable lnstr was significantly positive, and the regression model coefficient was a constituent of the direct effect, signifying that the regional

employment structure can be further optimized if a larger proportion of the regional industrial structure is inclined toward the tertiary industry. The regression coefficient of W.lnstr was significantly positive, indicating that the upgradations of the industrial structure in the neighboring regions will alter the employment structure of the given region. This can be attributed to the high mobility of the production labor. The improvement of the industrial structure level in the neighboring areas will attract the labor from the surrounding regions for employment in the manufacturing or secondary industries, thereby transforming the regional employment structure. Although an increase in the GDP per capita will further promote the optimization of the employment structure at the regional industry level, it will limit the optimization of the employment structure at the industry level in the neighboring regions. This is potentially because an adequate degree of regional development can attract capital inflows and government investment, which will consequently inspire the renewal and transformation of enterprises and promote the optimization of employment structure. However, if the developmental activities of this region utilize an extensive amount of resources, the development of the neighboring areas will be restricted and their employment structure will be negatively impacted. Notably, an increase in the labor supply capacity significantly inhibits the employment structure at the industrial level. Thus, if the extent of labor supply is greater than that of the demand in the region, a scenario of labor surplus will occur. As market competition has intensified and the numerous jobs may not be required, unemployment continues to increase. The rise in the rate will inhibit the development of the employment structure.

**Direct effect, indirect effect, and total effect analysis.** As the traditional spatial spillover effect analysis method poses certain defects, the present analysis was conducted from the perspective of indirect effects [22]. Specifically, the direct effect considers that the DE development in a given region influences the employment structure in the same region or that of the neighboring regions. Generally, indirect effects aid in assessing the impact on the employment structure of the given regions caused by the level of DE development in the neighboring regions [32]. The total effect is expressed as the sum of direct and indirect effects.

Herein, the spatial adjacency distance weight matrix was used to test Eq (2), and the estimation results are listed in Table 5.

In terms of direct effects, at the 5% and 1% significance levels, the DE development can directly influence the optimization of the employment structure at the industry level. Notably,

**Table 5. Regression results of direct effects, indirect effects, and total effects.**

| Variable | Model 3 | | | Model 4 | | |
|---|---|---|---|---|---|---|
| | lnis | | | Lnies | | |
| | **Direct Effect** | **Indirect Effect** | **Total Effect** | **Direct Effect** | **Indirect Effect** | **Total Effect** |
| lnde | 0.049** | 0.149** | 0.199** | 0.129 *** | 0.628 *** | 0.756 *** |
| | (2.45) | (1.86) | (2.08) | (6.33) | (3.69) | (4.49) |
| lnstr | 0.470 *** | 1.435 *** | 1.960 *** | 0.567 *** | 0.768 *** | 1.334 *** |
| | (16.02) | (2.69) | (3.54) | (21.52) | (3.87) | (6.76) |
| lngov | −0.483 *** | −1.473 *** | −1.956 *** | −0.558 *** | 0.955 *** | 0.397 |
| | (−14.59) | (−2.68) | (−3.52) | (−19.03) | (3.85) | (1.58) |
| lnred | 0.045 * | 0.137 | 0.181 | 0.075 *** | 0.062 | 0.137 |
| | (1.70) | (1.39) | (1.50) | (3.10) | (0.54) | (1.23) |
| lncst | 0.040 *** | 0.122** | 0.163** | −0.084 *** | −0.148 * | −0.232 *** |
| | (3.61) | (2.10) | (2.51) | (−8.23) | (−1.81) | (−2.84) |

Note: Values in parentheses present the *t*-statistics; the province- and year-fixed effects were not listed for brevity.

the regression coefficients describing the industrial structure, level of regional economic development, and CST were positive and were significant under the significance level of 1%, 10%, and 1% respectively, which indicates the direct influence of these factors toward promoting the employment structure of the region. Interestingly, governmental financial investment creates a significantly negative impact (1% significance) on the employment structure, because policy support for agricultural development increases the flow of labor and capital toward primary industries. Contrary to the employment structure of the labor force, the employment structure is suppressed, which is consistent with the aforementioned findings.

The significance level of DE development considering indirect effects is consistent with that of the direct effects, which can indirectly promote the optimization of the employment structure prevailing in the neighboring regions, with an evident spatial spillover effect relative to the level of DE development. At the 1% significance level, the industrial structure significantly promotes the optimization of the employment structure at all levels in the neighboring regions. As such, the impact of CST on the employment structure at all levels is consistent with previous reports. The regression coefficient of government financial support continues to inhibit the optimization of the employment structure at the industry level in neighboring regions at 1% significance level. Nonetheless, it can increase the number of secondary industries and promote the optimization of employment structure at the industry level. The regression coefficient reflecting the impact of GDP per capita on the provincial employment structure was not significant. Therefore, the level of regional economic development exerts did not evidently influence the level of employment structure in the neighboring regions.

In terms of the total effect, the DE development in a given region promoted the optimization and upgradation of the employment structure at all levels within the region (5% significance level) as well as improve the employment structure in the neighboring regions at 1% significance levels. Moreover, a significant positive spillover effect was observed by the original hypothesis H1. Furthermore, the industrial structure significantly (1% significance level) promoted the optimization and upgradation of the employment structure in the given region as well as the neighboring regions. The present findings revealed that the government's investments exerted a direct inhibitory effect (1% significance level) on the regional employment structure at the industry level. In addition, it hinders the DE development in the neighboring regions, and the value of the corresponding coefficient is negative because the total effect is equal to the sum of the direct and indirect effects. The impact of GDP per capita in the province on the employment structure at all levels is consistent with previous findings.

Generally, the impact of DE development level on employment structure is significantly positive (5% significance level), and the indirect effect (model regression coefficient: 0.149 and 0.628) was stronger than the direct effect (model regression coefficient: 0.049 and 0.129), indicating that the impact of DE development on the employment structure of the neighboring regions was greater than the impact on the region [33]. This can be attributed to the promotion primarily caused by the diffusion of the spillover effects, whereas its own promotion is weak, and the digital characteristics and agglomeration characteristics of the DE reduce the cost of diffusion and increase the cost of self-generation. Consequently, the spillover effect in the neighboring regions is strengthened and intensified.

## Mediation effect analysis

Under the framework of the above analysis, the intermediary effect of the degree of economic agglomeration and the level of innovation and entrepreneurship was tested from the spatial perspective, and the role path of the DE for promoting the optimization of the employment

**Table 6. Regression results of intermediary effect of degree of economic agglomeration.**

| Variable | lnden | | | | | | | | |
|---|---|---|---|---|---|---|---|---|---|
| | Model 5 | | | Model 6 | | | Model 7 | | |
| | | | | Industry Level | | | Industry Excess Level | | |
| | Direct Effect | Indirect Effect | Total Effect | Direct Effect | Indirect Effect | Total Effect | Direct Effect | Indirect Effect | Total Effect |
| lnde | 0.305 *** | 1.152 *** | 1.458 *** | 0.046 *** | 0.102** | 0.149** | 0.117 *** | 0.534 *** | 0.651 *** |
| | (9.00) | (2.98) | (3.65) | (2.42) | (1.97) | (2.19) | (5.95) | (3.20) | (3.93) |
| lnden | | | | 0.033 *** | 0.075** | 0.109 *** | 0.027** | 0.025 | 0.051 |
| | | | | (2.85) | (2.36) | (2.67) | (2.48) | (0.53) | (1.12) |
| lnstr | 0.797 *** | 3.002 *** | 3.799 *** | 0.449 *** | 1.022 *** | 1.471 *** | 0.553 *** | 0.712 *** | 1.265 *** |
| | (16.04) | (3.13) | (3.93) | (14.39) | (3.02) | (4.21) | (20.08) | (3.78) | (6.89) |
| lngov | −0.773 *** | −2.915 *** | −3.689 *** | −0.498 *** | −1.134 *** | −1.632 *** | −0.539 *** | 0.974 *** | 0.435 * |
| | (−13.85) | (−3.07) | (−3.83) | (−16.91) | (−3.07) | (−4.33) | (−17.22) | (3.94) | (1.76) |
| lnred | −0.253 *** | −0.956 *** | −1.209 *** | 0.093 | 0.052** | 0.119 * | 0.084 *** | 0.116 | 0.201 * |
| | (−5.74) | (−2.62) | (−3.12) | (0.67) | (2.07) | (1.67) | (3.62) | (1.01) | (1.76) |
| lncst | 0.246 *** | 0.931 *** | 1.178 *** | 0.033 *** | 0.075** | 0.108** | 0.089 *** | −0.142 * | −0.231 *** |
| | (13.14) | (3.03) | (3.77) | (2.90) | (2.00) | (2.32) | (−8.22) | (−1.80) | (−2.96) |

structure was verified by upgrading the level of innovation and entrepreneurship as well as the degree of economic agglomeration.

As listed in Tables 5–7, models 1–5 and 8 represent the regression results of the spatial Durbin model under double-fixed effects; the test steps and the corresponding results are stated as follows.

First, the effect of the DE development level on the employment structure was tested using Eq (2). As presented under Model 1 in Table 5, the regression coefficient of the total effect of the DE on the employment structure at all levels was positive at the 5% level of significance, i.e., the total effect path *c* is significant. Second, the influence of DE level on economic agglomeration degree and innovation and entrepreneurship level was tested using Eq (3), as presented under models 5 and 8 in Tables 6 and 7. The results revealed that the regression coefficient of

**Table 7. Regression results of mediating effect of innovation and entrepreneurship level.**

| Variable | lninn | | | | | | | | |
|---|---|---|---|---|---|---|---|---|---|
| | Model 8 | | | Model 9 | | | Model 10 | | |
| | | | | Industry Level | | | Industry Excess Level | | |
| | Direct Effect | Indirect Effect | Total Effect | Direct Effect | Indirect Effect | Total Effect | Direct Effect | Indirect Effect | Total Effect |
| lnde | 0.062 *** | 0.089 *** | 0.151 *** | −0.009 | −0.013 | −0.022 | 0.111 *** | 0.663 *** | 0.774 *** |
| | (13.69) | (4.68) | (7.17) | (−0.47) | (−0.44) | (−0.46) | 5.50 | 3.76 | 4.42 |
| lninn | | | | 0.982 *** | 1.334 *** | 2.316 *** | 0.973 *** | −1.735 | −0.761 |
| | | | | (10.99) | (3.54) | (5.81) | 2.86 | −0.85 | −0.38 |
| lnstr | 0.004 | 0.006 | 0.010 | 0.461 *** | 0.629 *** | 1.089 *** | 0.568 *** | 0.779 *** | 1.347 *** |
| | (0.64) | (0.59) | (0.61) | (16.10) | (3.47) | (5.77) | 21.69 | 4.20 | 7.45 |
| lngov | −0.037 *** | −0.053 *** | −0.089 *** | −0.449 *** | −0.613 *** | −1.062 *** | −0.554 *** | 1.046 *** | 0.492** |
| | (−4.93) | (−3.66) | (−4.43) | (−13.24) | (−3.49) | (−5.76) | −17.92 | 4.01 | 1.88 |
| lnred | 0.017 *** | 0.024 ** | 0.041 *** | 0.021 | 0.029 | 0.050 | 0.072 *** | 0.088 | 0.160 |
| | (2.83) | (2.47) | (2.69) | (0.87) | (0.82) | (0.85) | 3.16 | 0.77 | 1.43 |
| lncst | 0.056 *** | 0.082 * * | 0.139 *** | −0.014 | −0.019 | −0.033 | −0.094 *** | −0.103 | −0.197 * |
| | (22.64) | (4.90) | (7.98) | (−1.13) | (−0.03) | (−1.09) | −8.41 | −1.02 | −1.96 |

the total effect of DE level on the economic agglomeration degree and the innovation and entrepreneurship level is significantly positive (1% significance level), i.e., the DE development level positively influences the agglomeration of enterprises, funds, technologies, and capital. In particular, the degree of regional economic agglomeration is improved through DE development relying on big data, the internet, and other high-tech industrial applications, which creates innovation in working methods and tools as well as ushers several new industries. Thus, to promote the level of regional innovation and entrepreneurship, path *a* is significant. The regression coefficient of indirect the effects was significantly positive at the 1% significance level, indicating a positive spatial spillover effect of economic agglomeration and innovation and entrepreneurship on the intermediary effect of the employment structure. Third, the influence of DE development level, economic agglomeration degree, and innovation and entrepreneurship index on the employment structure level was determined using Eq (4), presented under models 6 and 7 in Table 6 and models 9–10 in Table 7. The results for model 6 indicated that the regression coefficients representing the total effect of the DE level and the economic agglomeration degree on employment structure were significantly positive at 5% and 1% significance levels, respectively, i.e., both paths B and *c'* were significant. Thus, the degree of economic agglomeration exerts a partial mediating effect, and the proportion of the mediating effect to the total effect can be evaluated as follows: $(1.458 \times 0.109)/0.199 = 79.86\%$. In model 9, the level of DE development exerts an insignificant reverse inhibitory effect on the employment structure, i.e., $\beta'_1$ in Eq (4) is not significant but poses a complete mediating effect on the employment structure. After conducting bootstrap tests for models 7 and 9, paths AB and *c'* of the two intermediary variables reflecting the employment structure at the industrial level were significant. This finding indicates a partial intermediary effect, with the same significance level for the symbols. The proportion of the intermediary effect of the level of innovation and entrepreneurship and the degree of economic agglomeration to the total effect can be calculated as follows: $(0.151 \times 0.761)/0.756 = 15.2$ and $(1.458 \times 0.051)/0.756 = 9.83$ respectively. Therefore, compared to the degree of economic agglomeration at all levels, the level of innovation and entrepreneurship posed a stronger intermediary effect on the employment structure. Therefore, hypothesis H2 is proved.

## Robustness test

**Tail reduction and elimination of municipalities.** The robustness of the model was tested by shrinking the tail of the explanatory and intermediary variables of the model, replacing the spatial weight matrix [34] of the model, and eliminating the municipality-level data [35]; the results are listed in Tables 8 and 9.

The level of DE development, degree of economic agglomeration, and the level of innovation and entrepreneurship in each city yielded extreme values of individual city data owing to various special reasons, thus affecting the regression results of the model. Therefore, both the core explanatory variables and the intermediary variables in the model are reduced, i.e., 1% before and after removing the core explanatory variables and intermediary variables, and the re-estimated results are listed in models 11–18 in Tables 8 and 9. Among them, the results of models 11 and 13 show that the three major effects of the development level of digital economy are positive and still significant at the level of 1%. The significance of the direct and indirect effects of economic agglomeration and innovation and entrepreneurship level remains unchanged. The analysis conclusions are highly consistent with the conclusions in Tables 6 and 7. The results of models 12 and 14 exhibited that the significance and positive and negative values of the DE development level remained unchanged, and the level of innovation and entrepreneurship remained significant, which validates the robustness of the results.

**Table 8. Tail-down regression results.**

| Variable | Tail reduction | | | | | | | | | | | |
|---|---|---|---|---|---|---|---|---|---|---|---|---|
| | Industry Level | | | | | | Industry Excess Level | | | | | |
| | Model 11 | | | Model 12 | | | Model 13 | | | Model 14 | | |
| | Direct Effect | Indirect Effect | Total Effect | Direct Effect | Indirect Effect | Total Effect | Direct Effect | Indirect Effect | Total Effect | Direct Effect | Indirect Effect | Total Effect |
| lnde | 0.064 *** | 0.138 *** | 0.202 *** | −0.002 | −0.003 | −0.005 | 0.126 *** | 0.567 *** | 0.693 *** | 0.113 *** | 0.734 *** | 0.847 *** |
| | (3.13) | (2.32) | (2.70) | (−0.09) | (−0.11) | (−0.10) | (6.23) | (3.20) | (3.93) | (5.37) | (3.94) | (4.58) |
| lnden | 0.036 *** | 0.076** | 0.111 *** | | | | 0.031 *** | 0.021 | 0.052 | | | |
| | (2.95) | (2.43) | (2.76) | | | | (2.77) | (0.48) | (1.17) | | | |
| lninn | | | | 1.057 *** | 1.356 *** | 2.413 *** | | | | 0.331 *** | −0.710 | −0.379 |
| | | | | (11.19) | (3.71) | (6.24) | | | | (3.81) | (−1.13) | (−0.60) |
| control variable | Control | | | Control | | | Control | | | Control | | |

The economic development across various cities in China is not uniform. For instance, regions with higher level of economic development have more mature internet infrastructure [35] and a relatively higher degree of DE development and economic agglomeration and the level of innovation and entrepreneurship. As such, these areas are predominantly characterized by Municipalities directly under the jurisdiction of the central government. Therefore, to address the possible endogeneity problems induced by reverse causality, the municipalities were discarded from the sample set of cities, and the results are listed in models 15–18 in Table 9. Among them, the regression coefficients of the three major effects of the DE development level in model 15 were all relatively smaller, and the significance level of the indirect and total effects diminished beyond significance. Nonetheless, the direct effect retained its 1% significance level, the regression results of each explanatory variable did not change significantly, and the regression results were robust.

**Replacement space weight matrix.** As the variations in spatial weight matrix induced deviations in the model regression results, this research used the spatial adjacency distance weight matrix to reflect the spatial relationship between things. To test the robustness of the model, the anti-distance square weight matrix W03 was employed instead of the spatial weight matrix. The re-estimation results of models 19–22 are summarized in Table 10. The regression

**Table 9. Regression results excluding municipalities.**

| Variable | Excluding Municipalities | | | | | | | | | | | |
|---|---|---|---|---|---|---|---|---|---|---|---|---|
| | Industry Level | | | | | | Industry Excess Level | | | | | |
| | Model 15 | | | Model 16 | | | Model 17 | | | Model 18 | | |
| | Direct Effect | Indirect Effe | Total Effect | Direct Effect | Indirect Effect | Total Effect | Direct Effect | Indirect Effect | Total Effect | Direct Effect | Indirect Effect | Total Effect |
| lnde | 0.036 *** | 0.080 | 0.117 | −0.016 | −0.021 | −0.037 | 0.151 *** | 0.540 *** | 0.691 *** | 0.145 *** | 0.696 *** | 0.841 *** |
| | (1.76) | (1.46) | (1.59) | (−0.77) | (−0.71) | (−0.74) | (7.08) | (3.27) | (4.26) | (6.49) | (3.76) | (4.61) |
| lnden | 0.037 *** | 0.081 *** | 0.118 *** | | | | 0.030 *** | 0.030 | 0.060 | | | |
| | (3.10) | (2.31) | (2.71) | | | | (2.82) | 0.64 | (1.31) | | | |
| lninn | | | | 0.987 *** | 1.264 *** | 2.252 *** | | | | 0.218 *** | −0.629 | −0.411 |
| | | | | (10.91) | (3.55) | (5.94) | | | | 2.61 | −1.14 | −0.75 |
| control variable | Control | | | Control | | | Control | | | Control | | |

Table 10. Regression results for replacing model matrix.

| Variable name | Replace spatial weight matrix | | | |
|---|---|---|---|---|
| | Industry Level | | Industry Excess Level | |
| | Model 19 | Model 20 | Model 21 | Model 22 |
| lnde | 0.056 *** | 0.003 | 0.115 *** | 0.109 *** |
| | (2.77) | (0.14) | (5.67) | (5.22) |
| lnden | 0.068 *** | | 0.023** | |
| | (5.93) | | (2.32) | |
| lninn | | 1.172 *** | | 0.217 *** |
| | | (13.79) | | (2.80) |
| control variable | Control | Control | Control | Control |

results of each variable were consistent with the earlier results and did not vary significantly. The DE development level can still significantly promote regional economic agglomeration, improve the level of innovation and entrepreneurship, and increase the number of manufacturing and secondary industries, thereby improving the employment structure. The regression results are stable.

## Heterogeneity analysis

**Sample analysis of cities in different regions.** The economic development across China's eastern and central regions vary significantly, with gradient characteristics in the level of employment structure [36]. Considering this information, the 283 prefecture-level cities in China were categorized into three segments according to the standard of the National Bureau of Statistics, and the spatial Durbin model was used to analyze [37] and the heterogeneity of each region. The specific empirical test results are listed in Table 11 and Figs 2 and 3.

The study unraveled that the DE development poses a significantly positive effect (1% significance level) on the employment structure of the industry in the eastern region of China. However, the industry level does not significantly affect the employment structure, possibly because of the saturation of the employed population in the eastern region. These findings imply the imminent need for emerging industries to distribute their labor force, which can be facilitated by the new employment and entrepreneurial opportunities resulting from the development of the DE. Therefore, the employment structure is more inclined to transition toward the tertiary industry, which will aid in optimizing the employment structure. Notably, the aforementioned factors did not pose a significant effect on the employment structure of the central region, which reduced the proportion of employment in secondary industries. This can

Table 11. Results of heterogeneity analysis in various regions.

| Variable name | Model 23 | | Model 24 | | Model 25 | |
|---|---|---|---|---|---|---|
| | lnis | lnies | lnis | lnies | lnis | lnies |
| | Eastern Region | | Central Region | | Western Region | |
| lnde | 0.138 *** | 0.50 | 0.080 | -0.027 | 0.057 | 0.207 *** |
| | (4.84) | (1.57) | (1.58) | (-0.71) | (1.14) | (4.74) |
| W.lnde | 2.771 *** | 1.047** | 0.187 | 2.924 *** | 1.364** | −0.219 |
| | (6.17) | (2.27) | (0.42) | (3.77) | (2.02) | (−0.37) |
| Number of observations | 900 | 900 | 882 | 882 | 765 | 765 |

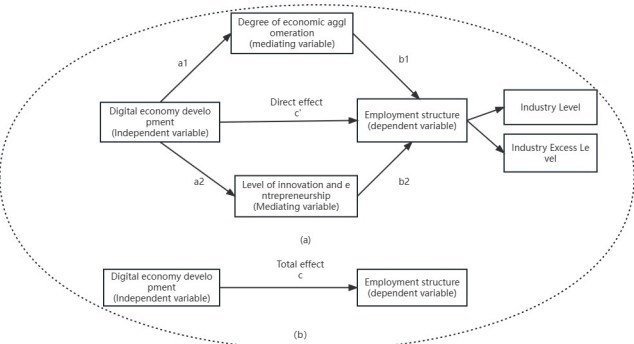

**Fig 2. Mediation effect model.**

be ascribed to the relatively solid industrial foundation in the central region, because of which the traditional manufacturing industry chain is relatively mature, and the innovation and application of the DE is relatively slow. Thus, less number of employees are affected by the DE development. In contrast, the western region poses a negative rather than significant impact on the employment structure of the industry prevalent in this region. However, it can significantly increase the employment population of the secondary industry, which is closely related to the dearth of resources in the western region. Therefore, hypothesis H3 is proved.

Based on a spatial perspective, the impact of the DE in the eastern region on the employment structure is significantly positive at the 1% significance level, because the DE encompasses a wide range of development, Certain digital economy enterprises will promote the development of related industries and form the upstream and downstream effects of the industrial chain. Secondary industries such as manufacturing in surrounding cities will develop accordingly with the increasing number of digital economy enterprises, and consequently, the number of employees is expected to increase. Although the employment structure of the western region positively affects the industry-level employment structure of neighboring regions at the 10% significance level, it does not create significant positive spillover effects to the industrial-level employment structure of the western region, which may be caused by the concentration of employment opportunities in the region to urban centers and the sinking of traditional industries. The spatial lag in the central region was not significant, indicating that the effect of DE on the industry-level employment structure requires further clarification for the cities located in the central region [37].

**Heterogeneity analysis of various city sizes.** Based on the research of Zhao et al. (2022) [38], this paper analyzed the size of cities according to the urban population, wherein the research sample included both large, medium, and small cities. The spatial Dubin model with double-fixed effects is used for empirical analysis of the large, medium, and small cities, and the test results are listed in Table 12 and Fig 4.

The results demonstrated that the level of DE development in small-and medium-sized towns in China posed a significantly positive effect (1% significance level) on the industry-level employment structure, where as it posed a significantly negative effect in large-scale cities. As large cities contain abundant resources with mature digital infrastructure, the level of DE development does not evidently impact the employment structure therein. Moreover, the reduction in marginal utility renders the employment structure as excessively biased toward a certain industry, thereby deteriorating the employment structure. In small-and medium-sized cities, the population size is limited and the digital foundation is weak. Therefore, the

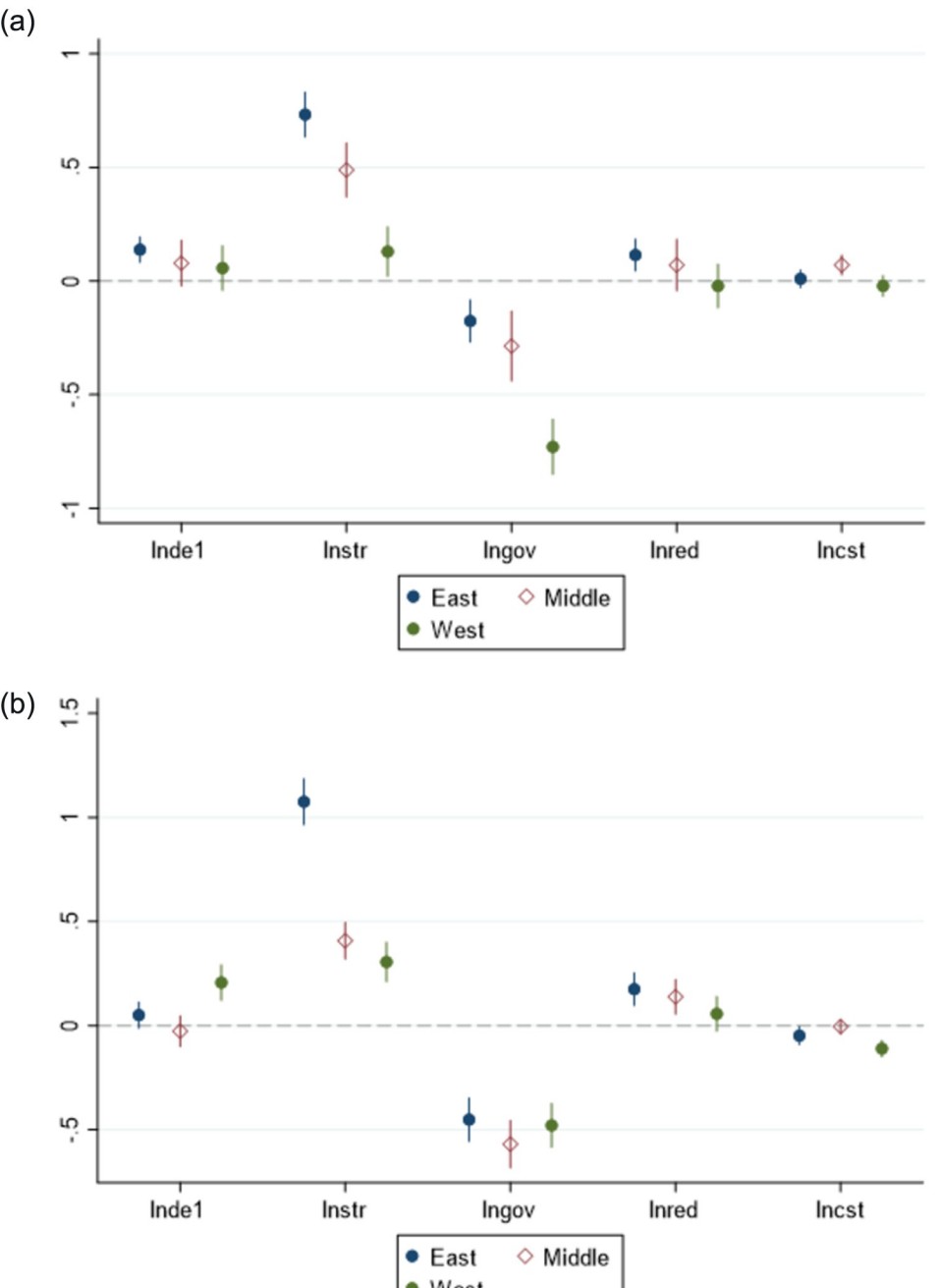

**Fig 3. Industry-level employment structure.** (a) lnis (b) lnies.

development of digital economy can significantly promote the optimization of employment structure. From a spatial perspective, small-sized cities were significantly positive at the 1% significance level, indicating that the improvement of the digital economy of similar cities will significantly optimize the employment structure of the region. The spatial lag of medium-sized cities and large-sized cities was not significant, implying that the effect of DE on the employment structure is still unclear. In the industrial-level employment structure, small- and

**Table 12. Results of heterogeneity analysis for varying city sizes.**

| Variable name | Model 26 | | Model 27 | | Model 28 | |
|---|---|---|---|---|---|---|
| | lnis | lnies | lnis | lnies | lnis | lnies |
| | small city | | medium-sized city | | large cities | |
| lnde | 0.324 *** | 0.285 *** | 0.194 *** | 0.110** | -0.187** | -0.265** |
| | (9.26) | (9.18) | (5.40) | (2.73) | (-2.46) | (-1.49) |
| W.lnde | 0.914 *** | 0.475** | 3.024 *** | 1.745 * | -0.606 | -0.470 |
| | (3.29) | (1.97) | (3.77) | (1.95) | (-1.52) | (0.13) |
| Number of observations | 1620 | 1620 | 801 | 801 | 126 | 126 |

medium-sized cities increase the number of employment in the region as well as increase the number of personnel employed in secondary industries functioning in neighboring areas, which optimizes the employment structure [39].

## Conclusions and recommendations

### Major conclusions

Using the panel data of 283 cities in China from 2011 to 2019, this study constructed a spatial Dubin model considering double fixation and intermediary effects and conducted an empirical analysis to characterize the relationship between the development of DE and the employment structure. The heterogeneity analysis was performed from two dimensions of varying regions and city sizes. The results indicated the following inferences: first, overall, the level of DE development in China can significantly optimize the employment structure of China, and after a series of robustness tests and heterogeneity analysis, the conclusion is still valid. Second, from the path of the digital economy acting on the employment structure, the degree of economic agglomeration and the level of innovation and entrepreneurship exert a significant intermediary effect on optimizing the employment structure, which was stronger than the degree of economic agglomeration. In particular, the degree of economic agglomeration poses a partial intermediary effect on the employment structure, with the intermediary effect accounting for 79.86% and 9.83% of the total effect, respectively. The digital economy significantly stimulates the vitality of urban innovation and entrepreneurship, creates more employment opportunities, and exerts a full intermediary effect and a partial intermediary effect of 15.2% on the employment structure at the urban and rural industry levels. Third, based on the heterogeneous effect of the digital economy on the urban employment structure, the development of the digital economy has significantly improved the employment structure of the major cities in the east, inhibited that in the central region, and affected that in the western region. Although the impact of DE is not entirely clarified, it can optimize the employment structure of small- and medium-sized cities; however, it significantly inhibits the improvement of the employment structure of large-scale cities.

### Policy recommendations

The aforementioned conclusions indicate that the development of the digital economy promotes the optimization of the employment structure and enhances this promotion through the positive intermediary effect of the degree of economic agglomeration and the level of innovation and entrepreneurship. To this end, this paper proposes the following suggestions to effectively improve the employment structure and quality of employment in China.

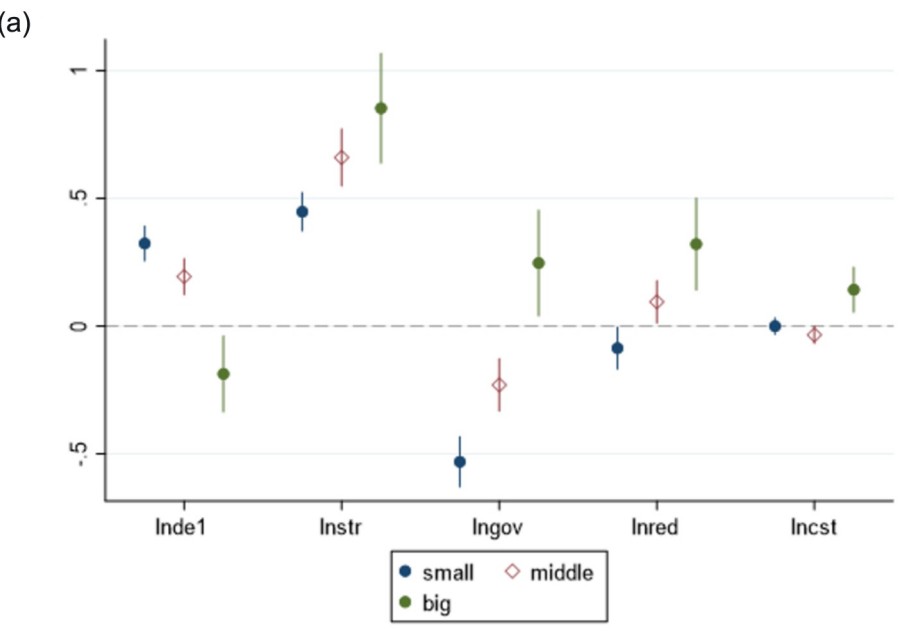

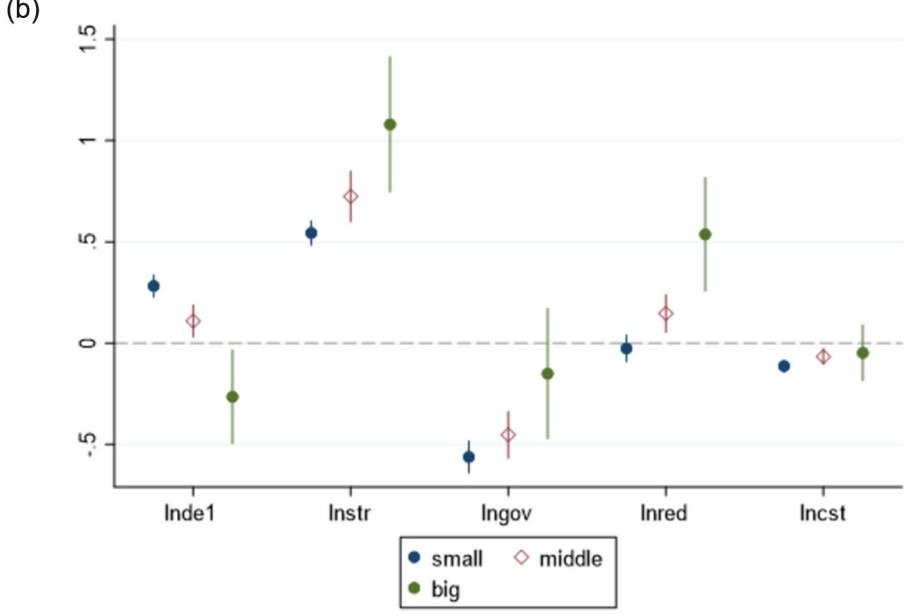

**Fig 4. Industry-level employment structure.** (a) lnis (b) lnies.

First, the rigorous development of the digital economy is required for optimizing the employment structure. Accordingly, attracting and encouraging the development of digital enterprises is one of the vital measures for optimizing the employment structure. Thus, the government can formulate certain supporting policies such as providing tax incentives for digital enterprises, venture capital, subsidized rent, which will attract more digital enterprises and provide broader opportunities and development prospects for the job market.

Second, the system construction must be improved with strengthened policy control. The government can employ policy means to focus on the development of digital industries,

increase investment in digital technology innovation, provide entrepreneurial training, and various additional supports to cultivate talents with digital technology and innovation capabilities, which will expand the job market as well.

Third, differentiated development policies must be implemented such that the development of the DE elements provides a reasonable distribution of employment. According to the resources, the human capital and industrial advantages of China's eastern, central, and western regions are distributed with relation to economic development. Thus, we should develop relevant industries according to the local conditions for improving the local employment structure. On the one hand, the layout of the digital economy must be strengthened in the central and western regions along with guided and optimized regional development to ensure marginal differences between regional development. On the other hand, the development of small- and medium-sized cities must be prioritized to ensure the fairness of the distribution of data resources among cities. These policies will ensure greater employment in small-sized towns.

## Supporting information

**S1 Data.**
(ZIP)

## Acknowledgments

We would like to thank the reviewers for providing professional comments on the manuscript.

## Author Contributions

**Data curation:** Lu Lu Zhou.

**Methodology:** Yang Lu.

**Software:** Yang Lu.

**Writing – original draft:** Lu Lu Zhou.

**Writing – review & editing:** Lu Lu Zhou.

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
