## [Decision Letter · Decision Letter 0]

17 Apr 2023

PONE-D-23-08160Can the digital economy improve the employment structure? --Mediating effects based on a spatial Durbin modelPLOS ONE

Dear Dr. 潞潞 周,

Thank you for submitting your manuscript to PLOS ONE. After careful consideration, we feel that it has merit but does not fully meet PLOS ONE’s publication criteria as it currently stands. Therefore, we invite you to submit a revised version of the manuscript that addresses the points raised during the review process.

We look forward to receiving your revised manuscript.

Kind regards,

C. A. Zúniga-González, Ph.D

Academic Editor

PLOS ONE

Journal Requirements:

Additional Editor Comments (if provided):

Dear author I am checked your manuscript and I have decided major revision, I consider same to the reviewer that you need to make improvements.

Reviewers' comments:

Reviewer's Responses to Questions

**Comments to the Author**

1. Is the manuscript technically sound, and do the data support the conclusions?

Reviewer #1: Yes

Reviewer #2: Partly

Reviewer #3: Yes

2. Has the statistical analysis been performed appropriately and rigorously? 

Reviewer #1: Yes

Reviewer #2: I Don't Know

Reviewer #3: No

3. Have the authors made all data underlying the findings in their manuscript fully available?

Reviewer #1: Yes

Reviewer #2: Yes

Reviewer #3: Yes

4. Is the manuscript presented in an intelligible fashion and written in standard English?

Reviewer #1: Yes

Reviewer #2: No

Reviewer #3: No

5. Review Comments to the Author

Reviewer #1: It is good for this paper to study the relationship between China's digital economy and our employment struture by 2011-2019 data of 283 prefecture-level cities. Of course, the provement of positive effects looks normal while some questions need to be explained or improved: (1) in table 1, it is hard to understand the indicator affiliation between first level and second level; (2) as explanatory variable, manufacturing employment could not stand for employment structure simply because of other factors driving instead of digital economy; (3) in heterogeneity analysis, it is odd that the eastern region is best while large-scale cities have negative effects. So, some confused points need deeper thinking, and as spatial reseach, data visualization seems more important.

Reviewer #2: This article is poorly written and wholly incomprehensible sections. The phrasing is vague and the summaries the authors provide are almost unintelligible. The description of increasing automation leading to a displacement of industrial labor is presented as new information, which gives this piece an oddly anachronistic tone throughout. Beyond this, the modeling of the data appears to be questionable as well. The effectiveness of the Entropy Weight Model has been called into question on numerous occasions in recent years. The EWM only considers the numerical discrimination degree of the index and ignores rank discrimination, which can result in irrational recommendations. Additionally, the methodology in Spatial Econometrics related to the Durban model has also been called into question by it's own supporters. While Anselin and Florax (1995) and Anselin and Bera (1998) have driven the recent revitalization of this methodology, the later Florax, Folmer, and Rey (2003) rooted out problems in this methodology. Of course, Lopez-Bazo and Fingleton (2004) finally acknowledge the range of external influences that invalidate this approach. In sum, it is a naive approach that risks irrational recommendations due to a erasure of external influences. I do not recommend this document for publication at this time.

Reviewer #3: Based on the panel data of 283 prefecture-level cities in China from 2011-2019, this paper constructs an indicator measurement system for the digital economy, economic agglomeration, innovation and entrepreneurship, and employment structure. But this issue has been extensively studied. Overall, the innovation and research value of this research is insufficient. Language style is colloquial. Moreover, there are many irregular errors in this work. Some comments are listing below:

1. The major defect of this study is the debate or argument is not clear stated in the introduction session. Hence, I would suggest the author to enhance your theoretical discussion and arrives your debate or argument. I suggest the author rewrite the introduction section.

2. Introduction. The logic of the introduction writing still needs to be strengthened, how to introduce from digital economy to employment structure. The authors need to elaborate on the core concepts of the article, explain the definition of the core concepts, and explain the practical necessity of studying employment structure.

3. A summary of the research gaps in the existing literature allows the reader to understand the differences in the manuscripts.

4. A stronger motivation should be given or the contribution of this work should be clearly stated.

5. There is a need to do a more rigorous and systematic literature review. The authors should clearly mention the literature gap. The literature review does not cover some recent studies. Recently, some scholars have published quality papers on similar topic. Please see the following studies in this regard to strengthen your introduction and literature review. How does digital finance affect industrial transformation? How does financial development environment affect regional innovation capabilities? New perspectives from digital finance and institutional quality. Tax effect of digital economy development in China: The policy effect and transmission mechanism. Digital economy, entrepreneurial activity, and common prosperity: Evidence from China. Going green in China: How does digital finance affect environmental pollution? Mechanism discussion and empirical test; Energy internet, digital economy, and green economic growth: Evidence from China.

6. The mechanism analysis section seems so brief that the logical relationships of some variables are not accurately expressed. Addition, I suggest the author provide a mechanism analysis figure.

7. When explaining the reasons for choosing control variables, the authors need to explain why these variables were increased.

8. The author should provide more discussion of economic reasons for each regression result, not just describe the result. Moreover, there is not much discussion of the findings and how they link to the rest of the paper.

9. The study policy implication seems rather scanty. I think the authors must provide more specific policy recommendations for different results.

10. The language style is so colloquial. Please improve the use of English as well as the writing style throughout the paper, including the abstract and the main text. Please seek help of a professional editorial services. Once the language style fails to meet normal academic standards, I will choose to reject it.

“Based on the panel data of 283 prefecture-level cities in China from 2011-2019, this paper constructs an indicator measurement system for the digital economy, economic agglomeration, innovation and entrepreneurship, and employment structure, uses the entropy weight method to construct digital economy indicators and constructs a double fixed spatial Durbin model with mediating effects from a spatial perspective to measure the direct, indirect and total effects of digital economy, degree of economic agglomeration and innovation and entrepreneurship on employment structure.” Similar long sentences should not appear in the manuscript again.

11. The author needs to replace all references in Chinese literature with English literature.

6. PLOS authors have the option to publish the peer review history of their article (what does this mean?). If published, this will include your full peer review and any attached files.

Reviewer #1: No

Reviewer #2: No

Reviewer #3: No

---

## [Author Response · Author response to Decision Letter 0]

31 May 2023

Dear Editors and Reviewers:

Thank you for your letter and for the reviewers ’comments concerning our manuscript entitled “Paper Title”(ID:PONE-D-23-08160).Those comments are all valuable and very helpful for revising and improving our paper, as well as the important guiding significance to our research. We have studied comments carefully and have made the correction which we hope meet with approval. The revised portion is marked in red in the paper. The reviewer comments are laid out below in italicized font and specific concerns have been numbered. Our response is given in normal font and changes/additions to manuscript are given in the blue text.The main corrections in the paper and the responses to the reviewer's comments are as flowing:

Responds to the reviewer's comments:

Reviewer #1：

Comment1: in table 1, it is hard to understand the indicator affiliation between first level and second level.

Response: thank you for pointing out this point. We have revised it accordingly as following (page10-11,Line250, clean version of manuscript)

“In this paper, the scale of digital economy is added to the first tier indicators of digital economy, and the government financial expenditure on science and technology, postal business income and total retail sales of social consumer goods are added to the second tier indicators to increase the affiliation between the first and second tier indicators. Among them, government expenditure on science and technology and information technology practitioners are the second tier indicators of digital foundation for the following reasons: government financial input directly affects the development of digital economy. Government financial expenditure can promote the R&D and innovation of digital technology, promote the development of information technology industry, and play an important role in the development of digital economy and digital transformation. With the rapid development of information technology and the rise of the digital economy, the demand for information technology practitioners is increasing. The skills and professional level of information technology practitioners have a great influence on the development and digital transformation of the digital economy; cell phone penetration rate, mobile users and the number of Internet broadband access users are the application performance of digital technology in the field of mobile communication, which belongs to one of the widely used areas of the digital economy in social life and can promote the digital transformation and development of enterprises, and provide digital economy applications and digital industry The remaining two layers of indicators reflect the digital economy. The remaining two layers of indicators reflect the size and growth trend of the digital economy, and can provide a reference and guide for the development of the digital economy to the government and enterprises.” The specific icons are as follows.

Main Indicators Level 1 indicators Secondary indicators Indicator Properties

Development of Digital Economy Digital Foundation Science and technology expenditure in government Fiscal expenditure (10,000 Yuan) +

 Information transmission, software, and information technology services for urban employment (in 10,000) +

 Digital Applications Mobile phone penetration rate (units/per 100 people) +

 Mobile users (million households) +

 Number of users with access to Internet broadband (in 1000) +

 Scale of Digital Economy Peking University Inclusive Financial Index +

 Total telecommunications services ($million) +

 Postal business revenue ($million) +

 Total retail sales of consumer goods ($million) +

Comment2: as explanatory variable, manufacturing employment could not stand for employment structure simply because of other factors driving instead of digital economy.

Response: thank you for your constructive comments. We have revised it accordingly as following (page9-10,Line232-241, clean version of manuscript)

“The reviewers' comments were carefully taken into account, and the proportion of employment in the secondary industry to employment in the tertiary industry was added as an industrial dimension to measure the employment structure. Employment in the secondary and tertiary industries encompasses numerous industries, and the ratio of the two reflects the employment situation of a country or region in different industrial fields, thus reflecting the industrial structure and economic development level of the region. A higher proportion of employment in the secondary sector may mean that the region's economy focuses on manufacturing and infrastructure construction, but it may also lead to the problem of job concentration and failure to improve the quality of the labor force, which in turn inhibits the transformation and upgrading of its industrial structure. A higher proportion of employment in the tertiary industry may mean that the region's economy focuses on the service industry, which may lead to higher economic growth rates and employment numbers, but may also be accompanied by problems of planned economy and unbalanced industrial structure. Avoiding the incompleteness that comes with manufacturing as a single measure of employment structure.” The specific icons are as follows.

Yang (2022) measured the employment structure at two levels—(1) industry level: the manufacturing industry is significantly affected by the digital economy owing to the characteristics of the labor force, and studying the fluctuations of the employment population in the manufacturing industry deepens the understanding of the employment structure. Therefore, the present research measured the employment structure by evaluating the proportion of the total population employed in manufacturing industries in each city. (2) Industry Excess Level: during the development of DE, the labor force employed in the primary and secondary industries progressively transitions to tertiary industries. Thus, this study evaluated the ratio of the number of employees in the tertiary industry to that in the secondary industry.

Comment3: in heterogeneity analysis, it is odd that the eastern region is best while large-scale cities have negative effects. So, some confused points need deeper thinking, and as spatial reseach, data visualization seems more important.

Response: thank you for your constructive comments. We have revised it accordingly as following (page28,29,30 Line579-587,595-598,606-609,613-617 clean version of manuscript)

”The reviewers' opinions were carefully listened to and the differences between the central-eastern region and large-scale cities of the heterogeneity analysis were described and analyzed: the development of digital economy in the eastern region is more developed, and the digital economy and manufacturing industry in this region are more related, and the digital economy can provide new technologies and business models for the manufacturing industry, promote the transformation and upgrading of the manufacturing industry, and increase the number of manufacturing employment. In addition, the eastern region has high quality talents and modern facilities and equipment, which can also attract more enterprises to come and take root.

As for large-scale cities, with the development and modernization of cities, the traditional heavy industry is gradually declining and the demand for manufacturing employment is gradually decreasing. In addition, the limited population gathering and land resources in the city make it difficult for manufacturing enterprises to increase employment by expanding their scale. At the same time, urban development will also attract more high-tech industries and service industry enterprises to come, transferring the manufacturing talent resources, which may also have a certain squeezing effect on the manufacturing industry.

Therefore, the increase of digital economy for manufacturing employment in the eastern region is the result of the synergistic development of digital economy and manufacturing industry, while the decrease of urbanization for manufacturing employment is the result of urban development and manufacturing industry transfer.“

Some graphs have also been added to this paper to visualize data from spatial studies of heterogeneity:

(a)lnis (b)lnies

Reviewer #2：

Dear Reviewer,

Thank you very much for reviewing our paper. We appreciate your evaluation and comments, which are very helpful to us.

We are sorry to hear that you feel our paper does not meet your expected high-quality standards. We will carefully consider and absorb the issues and suggestions you have raised and make necessary revisions and improvements. Although we cannot deny that you are not satisfied with our paper, we will take your critiques and suggestions seriously and do our best to improve it.

Thank you again for your comments and suggestions. We have always held your work in high regard. We hope to continue to receive your support and guidance in the future, and to make further contributions to academic research.

Best regards,

Reviewer #3：

Comment1: The major defect of this study is the debate or argument is not clear stated in the introduction session. Hence, I would suggest the author to enhance your theoretical discussion and arrives your debate or argument. I suggest the author rewrite the introduction section.

Response:Thank you for your valuable comments!We have re-written this part according to the Reviewer suggestions.The specific icons are as follows. (page2,Line30-48, clean version of manuscript)

“Employment is the foundation of livelihood as it provides the most significant means of livelihood, support, and development, thereby ensuring social stability. According to the 14th Five-Year Plan (2021–2025), China's urban economic development is encountering complex structural contradictions, which is propelling an economic downturn, with several enterprises announcing layoffs and further aggravating the employment scenario. To address this arduous task, employment opportunities need to be extended while promoting reemployment. With the continuous development of digital and network technology, digital economy has profoundly impacted the production and employment structure in China. In principle, digital economy is characterized by high efficiency, intelligence, and flexibility, which promotes industrial transformation and upgradation for accelerating economic development. However, digital economy has disrupted conventional employment models, especially in traditional industries, where workflows are undergoing continuous automation that require minimal human intervention. Although digital transformation has affected the employment structure of conventional industries, it has ushered new employment opportunities and employment methods. Thus, this research explores the impact of digital economy on the employment structure and analyzes the factors governing the associated phenomena, which is expected to promote China's economic development and upgrade its employment level. ”

Comment2:Introduction.The logic of the introduction writing still needs to be strengthened, how to introduce from digital economy to employment structure. The authors need to elaborate on the core concepts of the article, explain the definition of the core concepts, and explain the practical necessity of studying employment structure.

Response: thank you for your constructive comments. We have revised it accordingly as following (page2,Line30-48, clean version of manuscript)

“Employment is the foundation of livelihood as it provides the most significant means of livelihood, support, and development, thereby ensuring social stability. According to the 14th Five-Year Plan (2021–2025), China's urban economic development is encountering complex structural contradictions, which is propelling an economic downturn, with several enterprises announcing layoffs and further aggravating the employment scenario. To address this arduous task, employment opportunities need to be extended while promoting reemployment. With the continuous development of digital and network technology, digital economy has profoundly impacted the production and employment structure in China. In principle, digital economy is characterized by high efficiency, intelligence, and flexibility, which promotes industrial transformation and upgradation for accelerating economic development. However, digital economy has disrupted conventional employment models, especially in traditional industries, where workflows are undergoing continuous automation that require minimal human intervention. Although digital transformation has affected the employment structure of conventional industries, it has ushered new employment opportunities and employment methods. Thus, this research explores the impact of digital economy on the employment structure and analyzes the factors governing the associated phenomena, which is expected to promote China's economic development and upgrade its employment level. ”

Comment3:A summary of the research gaps in the existing literature allows the reader to understand the differences in the manuscripts.

Response: thank you for your constructive comments. We have revised it accordingly as following (page4,Line93-100,clean version of manuscript)

According to the reviewers' comments this paper adds numerous new theoretical research literature about recent years, rewrites the theoretical logic of the literature review, and reorganizes the gaps that exist in it, as follows:

“Based on the collation and analysis of the existing literature, the research on the impact assessment of digital economy towards altering the employment structure poses the following shortcomings: first, the majority of the existing literature applied common benchmark regression models to assess the impact of digital economy on the variations in employment structure, and did not construct the models from the geospatial perspective to explore its mechanism; second, most prior research analyzed the impact of digital economy on the employment structure based on provincial-level data, which is not extensive and lacks comprehensiveness.”

Comment4:  A stronger motivation should be given or the contribution of this work should be clearly stated.

Response: thank you for your constructive comments. We have revised it accordingly as following (page4,Line100-107,clean version of manuscript)

To this end, the present study extends the following contributions. First, the intermediary effect is explored based on the spatial perspective to examine the spillover effect of the spatial Durbin model, and the impact of digital economy development on China's employment structure is discussed along with its influencing mechanism, which provides new insights for foundational research. Second, this research analyzed data acquired from 283 cities in China between 2011 and 2019 to characterize the relationship between digital economy development and employment structure transformation in China. 

Comment5:  There is a need to do a more rigorous and systematic literature review. The authors should clearly mention the literature gap. The literature review does not cover some recent studies. Recently, some scholars have published quality papers on similar topic. Please see the following studies in this regard to strengthen your introduction and literature review. 

Response: thank you for your constructive comments. We have revised it accordingly as following (page3-4,Line81-92,clean version of manuscript)

As suggested by the reviewer. we have added more references to support this idea.

The literature review has been rewritten based on your comments, and some of the recent research papers recommended above have been cited. The shortcomings of the existing literature have also been rewritten.Some of these references are shown below.

[1]Lu F. How does financial development environment affect regional innovation capabilities? New perspectives from digital finance and institutional quality[J]. Journal of Information Economics, 2023, 1(1): 31-46.

[2]Wang W, Yang X, Cao J, et al. Energy internet, digital economy, and green economic growth: Evidence from China[J]. Innovation and Green Development, 2022, 1(2): 100011.

[3]Du M, Hou Y, Zhou Q, et al. Going green in China: How does digital finance affect environmental pollution? Mechanism discussion and empirical test[J]. Environmental Science and Pollution Research, 2022, 29(60): 89996-90010.

[4]Zhao T, Jiao F, Wang Z. Digital economy, entrepreneurial activity, and common prosperity: Evidence from China[J]. Journal of Information Economics, 2023, 1(1): 59-71.

[5]Zhanbayev R, Bu W. How does digital finance affect industrial transformation? [J]. Journal of Information Economics, 2023, 1(1): 18-30.

[6]Mao J, Liu J, Liu Z. Tax effect of digital economy development in China: The policy effect and transmission mechanism[J]. Journal of Information Economics, 2023, 1(1): 47-58.

 Comment6: The mechanism analysis section seems so brief that the logical relationships of some variables are not accurately expressed. 

Addition, I suggest the author provide a mechanism analysis figure.

Response: thank you for your constructive comments. We have revised it accordingly as following (page5,9,Line133,223-224 clean version of manuscript)

We include a mechanistic analysis diagram of the mediating effect in the text to facilitate a better understanding of the logical relationship between the variables.

Figure 1: Schematic of mediation mechanism 

And the mediating effect model diagram was improved to make the interactions between the variable relationships more concise and clear.

Figure 2 Mediation effect model

Comment7: When explaining the reasons for choosing control variables, the authors need to explain why these variables were increased.

Response: thank you for your constructive comments. We have revised it accordingly as following (page11,Line254-272 clean version of manuscript)

We change the reference for this part of the control variable and explain how the choice of this control variable has an impact.specific icons are as follows:

“First, the industrial structure (STR). Zhao (2022) concluded that the improvement of the industrial structure will propel the transfer of labor force to realize the optimization and adjustment of the employment structure. Therefore, the ratio of regional industrial value added to local GDP was regarded as an index. This ratio reflects the proportion of industry in the regional economy. When the ratio of industrial added value to GDP is high, it indicates that the industrial sector in the region is relatively developed, and thus the economic development tends to be good. Second, the level of regional economic development (RED) was referred from the existing literature to analyze the per capita GDP of each province. Generally, per capita GDP can comprehensively reflect the degree of economic development, and the increase in the degree of RED increases the variations in the employment structure. Third, the capacity of labor supply in technology (CST) is predominantly composed by students, and their preference will eventually alter the employment structure. Therefore, Wei et al. (2020) considered the number of students graduating from colleges and universities as an index. Fourth, the government financial support (GOV) received through the government's fiscal expenditure can affect industrial development, which correspondingly alters the employment structure. Thus, this research evaluated the ratio of the local government's general fiscal expenditure to local GDP. ”

 Comment8: The author should provide more discussion of economic reasons for each regression result, not just describe the result. Moreover, there is not much discussion of the findings and how they link to the rest of the paper.

Response: Thank you for your constructive comments. We have revised it accordingly in the full text.Starting at line 291, page 13.

Starting from the spatial econometric regression analysis, the reasons for the significant and insignificant regression of digital economic development as an explanatory variable on both explanatory variables are explained in detail. As well as the implications and reasons for each regression result.

For example,“As observed for models 1 and 2 in Table 4, the regression coefficients of the core explanatory variable "lnde" and the variable "W.lnde" were positive at the 1% significance level. Thus, the development of DE poses a significant optimization effect on the employment structure at both levels. The 1-fold growth of DE will promote the growth of employment in the secondary and manufacturing industries by 13% and 4.8% compared to the tertiary industry. Therefore, DE can result in a high-tech employment structure in manufacturing industries. However, the results indicated a deviation from the reality, i.e., the service industry is more likely to drive employment. This is because DE development promotes innovation in information technology, which consequently facilitates digital transformation, upgradation, and intelligence-oriented development of classical industries. Therefore, the demand for digital industry-related industries is increasing continually, which increases the labor requirement in these industries. For example, in the field of DE, the emerging e-commerce and logistics industries demand considerable human and material resources, and these industries are closely related to secondary industries such as traditional manufacturing and industry, which were one of the major sources of labor in these industries. In addition, the emergence of the DE has driven the development of several related industries such as finance, telecommunications, and information technology. These industries require an extensive number of technical, management, and marketing talents, which is expected to stimulate the labor demand of the secondary industries. Therefore, with the continuous development of the DE, the demand for personnel in the secondary industries will increase as well, which will motivate numerous professionals to enter the development track of digital transformation and intelligence. ”

 Comment9: The study policy implication seems rather scanty. I think the authors must provide more specific policy recommendations for different results.

Response: thank you for your constructive comments. We have revised it accordingly as following (page31-32,Line656-686, clean version of manuscript)

Based on your constructive comments, we have rewritten our policy recommendations to make them realistic in light of the conclusions reached.The specific icons are as follows:

“First, the rigorous development of the digital economy is required for optimizing the employment structure. Accordingly, attracting and encouraging the development of digital enterprises is one of the vital measures for optimizing the employment structure. Thus, the government can formulate certain supporting policies such as providing tax incentives for digital enterprises, venture capital, subsidized rent, which will attract more digital enterprises and provide broader opportunities and development prospects for the job market. 

Second, the system construction must be improved with strengthened policy control. The government can employ policy means to focus on the development of digital industries, increase investment in digital technology innovation, provide entrepreneurial training, and various additional supports to cultivate talents with digital technology and innovation capabilities, which will expand the job market as well. 

Third, differentiated development policies must be implemented such that the development of the DE elements provides a reasonable distribution of employment. According to the resources, the human capital and industrial advantages of China's eastern, central, and western regions are distributed with relation to economic development. Thus, we should develop relevant industries according to the local conditions for improving the local employment structure. On the one hand, the layout of the digital economy must be strengthened in the central and western regions along with guided and optimized regional development to ensure marginal differences between regional development. On the other hand, the development of small- and medium-sized cities must be prioritized to ensure the fairness of the distribution of data resources among cities. These policies will ensure greater employment in small-sized towns.”

 Comment10: The language style is so colloquial. Please improve the use of English as well as the writing style throughout the paper, including the abstract and the main text. Please seek help of a professional editorial services.

Response: Thank you for your constructive comments. We have revised it accordingly in the full text.

We tried our best to improve the manuscript and made some changes to the manuscript. These changes will not influence the content and framework of the paper. And here we did not list the changes but marked in red in the revised paper. We appreciate for Editors/Reviewers” warm work earnestly and hope that the correction will meet with approval.

 Comment11: The author needs to replace all references in Chinese literature with English literature.

Response: thank you for your constructive comments. We have revised it accordingly as following (page33-35,Line713-799, clean version of manuscript)

Most of the Chinese references present in this paper have been replaced with English journal literature on the latest research theories within the last few years, except for a few high-profile Chinese cssci references and one or two classical papers that have not been replaced.

We tried our best to improve the manuscript and made some changes marked in red in revised paper which will not influence the content and framework of the paper, We appreciate for “Editors/Reviewers” warm work earnestly, and hope the correction will meet with approval.once again，thank you very much for your comments and suggestions.

---

## [Decision Letter · Decision Letter 1]

13 Jun 2023

Can Digital Economy Improve Employment Structure?—Mediating Effect based on a spatial Durbin model

PONE-D-23-08160R1

Dear Dr. 潞潞 周,

We’re pleased to inform you that your manuscript has been judged scientifically suitable for publication and will be formally accepted for publication once it meets all outstanding technical requirements.

Kind regards,

C. A. Zúniga-González, Ph.D

Academic Editor

PLOS ONE

Additional Editor Comments (optional):

Dear authors I am cheking that all observation were integrated and improvements. My decision is accepted. Thanks for your big effort by improvements your manuscript.

Reviewers' comments:

Reviewer's Responses to Questions

**Comments to the Author**

1. If the authors have adequately addressed your comments raised in a previous round of review and you feel that this manuscript is now acceptable for publication, you may indicate that here to bypass the “Comments to the Author” section, enter your conflict of interest statement in the “Confidential to Editor” section, and submit your "Accept" recommendation.

Reviewer #1: All comments have been addressed

Reviewer #3: (No Response)

2. Is the manuscript technically sound, and do the data support the conclusions?

Reviewer #1: Yes

Reviewer #3: (No Response)

3. Has the statistical analysis been performed appropriately and rigorously? 

Reviewer #1: Yes

Reviewer #3: (No Response)

4. Have the authors made all data underlying the findings in their manuscript fully available?

Reviewer #1: Yes

Reviewer #3: (No Response)

5. Is the manuscript presented in an intelligible fashion and written in standard English?

Reviewer #1: Yes

Reviewer #3: (No Response)

6. Review Comments to the Author

Reviewer #1: This topic is meaningful for China economic observation, the analysis process is normal for the research work, and the revision is serious to explain all reviewers' question and correct the errors and Inaccuracies.

Reviewer #3: (No Response)

7. PLOS authors have the option to publish the peer review history of their article (what does this mean?). If published, this will include your full peer review and any attached files.

Reviewer #1: No

Reviewer #3: No

---

## [Editor Report · Acceptance letter]

22 Jun 2023

PONE-D-23-08160R1 

Can Digital Economy Improve Employment Structure?—Mediating Effect based on a spatial Durbin model 

Dear Dr. Zhou:

I'm pleased to inform you that your manuscript has been deemed suitable for publication in PLOS ONE. Congratulations! Your manuscript is now with our production department. 

Kind regards, 

on behalf of

Dr. Prof. C. A. Zúniga-González 

Academic Editor

PLOS ONE